# Cardiovascular risk in aging adults with double deficits in social support: A gender-sensitive, cross-sectional analysis of the CLSA cohort

Annalijn I. Conklin[1,2,3]*, Abdollah Safari[4], Gerry Veenstra[5], Nadia A. Khan[2,6]

1 Faculty of Pharmaceutical Sciences, University of British Columbia, Vancouver, Canada, 2 Centre for Advancing Health Outcomes, Providence Health Care Research Institute, St. Paul's Hospital, Vancouver, Canada, 3 Edwin S.H. Leong Centre for Healthy Aging, Faculty of Medicine, University of British Columbia, Vancouver, Canada, 4 School of Mathematics, Statistics, and Computer Science, Faculty of Science, University of Tehran, Iran, 5 Department of Sociology, University of British Columbia, Vancouver, Canada, 6 Department of Medicine, University of British Columbia, Vancouver, Canada

* aconklin@mail.ubc.ca

## Abstract

### Background

Synergistic effects of diverse social supports (informational, tangible, emotional and belonging) on cardiovascular disease risk factors (CVRF), by gender, is unknown.

### Aim

To quantify gender differences in the singular and combined associations of four different forms of social support with cardiovascular disease risk factors (CVRF) in aging adults.

### Methods

Cross-sectional study of 28,779 adults (45−85 years) in the Canadian Longitudinal Study on Aging Comprehensive cohort (2011−15); independent variables were self-reported measures of informational, tangible, emotional and belonging support; dependent variables were clinically measured BMI, waist circumference and blood pressure. We used stratified multivariable linear and logistic regression with principal component regression with cross-product terms to post-estimate adjusted means and 95% CIs for combined associations.

### Results

All low-low support combinations were consistently associated with the highest adjusted mean BMI and WC levels among women. Adjusted mean BMI differences were largest among women with low informational and low tangible supports (27.95 kg/m$^2$ [27.93, 27.97]), compared to women with high informational and high

**Data availability statement:** Due to privacy and confidentiality requirements data can only be made available for researchers who meet the criteria for access. See Canadian Longitudinal Study on Aging (CLSA) https://www.clsa-elcv.ca/data-access for details.

**Funding:** This research was funded by the Canadian Institutes of Health Research Catalyst Grant for Secondary Analysis of the CLSA (#162987). AIC received Faculty of Pharmaceutical Sciences start-up funds for hardware and software. Funding for the Canadian Longitudinal Study on Aging (CLSA) is provided by the Government of Canada through the Canadian Institutes of Health Research (CIHR) under grant reference: LSA 94473 and the Canada Foundation for Innovation, as well as the following provinces, Newfoundland, Nova Scotia, Quebec, Ontario, Manitoba, Alberta and British Columbia. This research has been conducted using the CLSA Baseline Comprehensive Dataset version 4.0, under Application Number 19CA003. The CLSA is led by Drs. Parminder Raina, Christina Wolfson and Susan Kirkland. The funders had no role in study design, data collection and analysis, decision to publish, or preparation of the manuscript.

**Competing interests:** The authors have declared that no competing interests exist.

tangible supports (27.34 kg/m$^2$ [27.30, 27.38]). Similarly, the greatest difference in adjusted mean WC was seen among women with low informational and low emotional supports (88.69 cm [88.62, 88.76]) compared to the high-high combination (86.88 cm [86.75, 87.01]). Women with low availability of informational support, with or without deficits in a second support type, had the highest adjusted mean SBP levels (range: 119.94 to 119.95 mmHg). Among men, mean CVRFs were not consistently worse for combinations of dual deficits in social support. Results were null for DBP.

## Conclusion

Women with two deficits in social supports, particularly combinations with low informational support, showed worse CVRF measures than one social support deficit. Results indicated no antagonistic/synergistic effects of social support on CVRFs. Heart health care and prevention for aging women would benefit from ensuring informational support with other supports is available.

## Introduction

Both hypertension and obesity prevalence contribute to the global burden of disease and are strong physiological determinants of (un)healthy aging [1]. Excess adiposity is more prevalent among women than men in most countries and extreme classes of obesity have risen faster for women than men in Canada [2]. Similarly, elevated blood pressure (BP) is a vital sign of health and a major cardiovascular risk factor (CVRF) that is more prevalent among women than men in later life [3]. Blood pressure rises more with age among women likely due to sex/gender and age differences in the changing balance of vasodilating and vasoconstricting adrenergic receptor tone [4]. The annual costs of hypertension alone were expected to be $20.5 billion in 2020 in Canada [5]. Preventing and treating both CVRFs is therefore a public health and policy priority in Canada and elsewhere [6].

Social support is a functional aspect of social connections that is known to impact cardiac and metabolic health [7], and is directly associated with psychological wellbeing [8,9]. Higher perceived social support improved quality of life in cardiovascular patients in South Asia, especially combined with high self-efficacy [10]. Although high social support buffers against life stress (stress buffering theory), high social support can combine with high social undermining (i.e., efforts by group members to intentionally hinder another member's success) to negatively impact health [11]. Importantly, the health buffering role of supportive networks may also differ by gender [12]; for example, synergistic interactions of social support and job control on depression and insomnia appear stronger among employed women than among men [13]. Greater informational support and emotional support are separately linked to lower blood pressure more strongly for women than men [14]. In addition, four types of social support (tangible, belonging, informational and emotional) were independently associated with adiposity among women but not men [15].

As a multidimensional construct, social support is conventionally assessed using a summary index (e.g., MOS score) [16], and so the interplay of different social supports is seldom considered. Whether the absence of one type of social support can be mitigated by the presence of another type or whether double deficits in supports combine to show additive or synergistic effects for women or men is unknown. Thus, it remains an empirical question to determine if one gender is overall more vulnerable to poor CVRFs from double deficits in support, or if women's and men's vulnerability varies by which combination is specified..

This cross-sectional study is novel in addressing a knowledge gap on and whether and which combinations of different social supports affect CVRFs, and whether joint associations differ by gender. We hypothesized that dyadic combinations between the four types of social support (informational, tangible, emotional and belonging) would be synergistically associated with the CVRFs wherein low levels of two forms of support correspond to inordinately worse measures of CVRFs. We also hypothesized that these synergistic/amplifying effects would be more pronounced among women than among men. If results show that double deficits in social support produce mutually reinforcing synergistic effects on CVRFs, or that joint associations are gender-specific, then public health and social prescribing may need to simultaneously address multiple types of social support or target each gender, or both.

## Materials and methods

### Study design and population

We used baseline data (2011–15) from 30,097 predominantly White, non-institutionalised, middle-aged and older adults (45–85 years) in the population-based Canadian Longitudinal Study on Aging (CLSA) Comprehensive cohort—a stratified random sample of community-dwelling individuals residing within a 25–50 km of 11 specified data collection sites [17]. This cohort had self-reported questionnaire data on physical, social and psychological factors and clinically measured CVRF data from English- or French-speaking individuals (cohort details in S1 File). This cohort excluded residents of the three territories, federal First Nations reserves and other First Nations settlements in the provinces, remote regions, full-time members of the Canadian Forces, and individuals living in institutions or with cognitive impairment at the time of recruitment [18]. Our available sample included participants with complete data on perceived availability of social support, covariables, anthropometry and blood pressure (n = 28,779). All participants gave written informed consent and the study was approved by the *[blinded for review]* Behavioural Research Ethics Board (H19-00971). CLSA data were fully anonymized for research purposes and were accessed on 31-07-2019.

### CVD risk factors (CVRFs)

We used two adiposity measures (body mass index (kg/m2) and waist circumference (cm)) and two cardiovascular measures (systolic blood pressure (SBP) and diastolic blood pressure (DBP)) as important indicators of cardiovascular disease (CVD) risk, and established physiological determinants of ageing [19]. We used clinically measured standing height (m) and weight (Kg), collected by Seca 213 stadiometer and 140−10 Healthweigh Digital Physician Scale, to calculate body mass index (BMI, Kg/m2). Waist circumference (CM) was measured from halfway between the last rib and the iliac crest bone [18]. We used the last five of six measurements to calculate a mean value for SBP and DBP for each participant; clinical measures were taken with the BpTRU™ BPM200 Blood Pressure Monitor [18]. We also used clinically relevant secondary outcomes of general obesity (BMI ≥ 30 kg/m2), central obesity (WC ≥ 88 cm for women and WC ≥ 102 cm for men) and hypertension (≥ 140/90 mmHg (≥ 130/80 for those with diabetes), self-reported diagnosis or use of hypertension medication).

### Four types of social support

We used four specific types of social support (emotional, informational, tangible, and belonging) that an individual may experience [20–22]. For each CLSA question about how often each of the supports was available if the participant needed

it, participants self-reported five responses coded as 1 = none of the time, 2 = a little of the time, 3 = some of the time, 4 = most of the time or 5 = all of the time [18]. The CLSA social support questions were based on the validated 19-item, self-administered Medical Outcomes Study Social Support Survey (MOS-SSS), which has overall excellent internal reliability (Cronbach's alpha > 0.90) and good test-retest reliability (ICC ≥ 0.84) [16] and good reliability of each dimension (CFA > 0.90) [23]. Questions assessing *emotional* social support asked about having someone: to listen when needing to talk, to confide in about oneself or problems, to share their most private worries and fears with, to love and make them feel wanted and who shows love and affection, and who hugs them. Responses were summed to calculate a score for emotional support (range: 7–35). Questions on *informational* support included having someone to give advice about a crisis, to give information in order to help, to turn to for suggestions about how to deal with a personal problem, and someone whose advice is really wanted, resulting in a summary score with range of 4–20 points. Questions about *tangible* social support (score range: 4–20 points) concerned availability of help if confined to a bed, someone to take to the doctor, to prepare meals if unable to do so alone, or to help with daily chores if sick. And questions regarding *belonging* support (score range: 4–20 points) reflected having someone to have a good time with, to relax with, to do things to help get one's mind off things, to do something enjoyable with, and who understands one's problems. Details in S1 Table in S1 File.

## Covariables

We included a parsimonious set of covariables based on the literature, biological plausibility and specialized content knowledge: self-reported age and age squared, smoking status (ever/never), province and education (less than secondary school; secondary school; some post-secondary education including degree/diploma; university degree). Our selection of covariables was informed by existing literature on factors known to correlate with our exposure (social ties) and our outcome [24,25], and were not mediators on the causal pathway (e.g., health behaviours). Nevertheless, results were checked for robustness to other factors that might be potential confounders including chronic conditions, marital status, health behaviours, psychological factors, and reproductive factors (variable details in S1 File). Models of adiposity were also re-specified to include blood pressure (and vice versa).

## Statistical analysis

Descriptive statistics (mean (SD) or proportions) were used to summarize socio-demographic characteristics and crude levels of cardio-metabolic outcomes in relation to the four functional social ties. As the CLSA Comprehensive cohort survey weights represented only Canadian population residing within 25 km of CLSA's Data Collection Site they were not used here. Different statistical tests were used to measure the strength of relationships between the functional tie variables.

A sequence of multivariate linear and logistic regression models was used to capture the joint associations of the different functional ties with each cardiovascular outcome, by gender, in order to determine whether dual deficits precipitate additive or synergistic effects. Stratified multivariable regression models were used to report estimates separately for (cisgender) women and men. The models for each outcome simultaneously accounted for known confounders (see covariables), functional ties, and two-way interactions (cross-product term) between each pair of four functional ties. Due to the high multicollinearity between the functional ties and also their interactions (variance inflation factors > 5), we used principal component regression models to analyze their interactive associations [26] that involved four main steps: 1) Derive principal components (PCs) on correlated covariates (functional ties [4 terms] and all of their two-way interactions [6 cross-product terms]); 2) From the computed PCs, retain a subset that sufficiently explained data variation while mitigating multicollinearity issues; 3) Fit a regression model on selected PCs along with other independent cofounders; and 4) Back-transform the coefficients' estimate of the PCs in the regression model to the coefficient of original correlated predictors by using the PCs' scores. To determine the number of PCs to retain in Step 2, we used R-squared and root mean squared error of prediction (RMSEP) criteria [27]. PCs were selected based on their contribution to variance explained and their ability to minimize RMSEP, ensuring robust model performance while preventing overfitting. In most cases, only

the last one or two PCs were excluded from the final regression models. It is important to note that because each PC is a linear combination of the correlated covariates, all functional ties and their interactions were inherently included in every PCR model, regardless of the number of PCs retained in Step 2. S2 to S4 Tables in S1 File report the number of selected PCs used in all PCR models, including an example of PCA scores used in the linear model of BMI among men.

Adjusted mean levels and 95% confidence intervals (CI95) of continuous outcomes (WC, BMI, SBP and DBP) were calculated based on post-estimation analysis for women and men and displayed visually for ease of interpretation of additive or multiplicative effects. Adjusted means of different combinations of levels of each functional tie pair were also compared using Tukey adjusted p-values for multiple testing. Sensitivity analyses were used to test the robustness of the joint associations to additional confounding in separate models further conditioned on chronic disease, marital status, health behaviours, psychological factors and reproductive status (covariable details in S1 File). Analyses were carried out in R (a language and environment for statistical computing. R Foundation for Statistical Computing, Vienna, Austria, 2020).

## Results

Our sample was 50.8% women with a mean age of 63 years (SD 10) (Table 1). Educational differences were seen between those with high versus low social support of each type. A higher proportion of Canadian adults with low social support were generally obese, centrally obese or had hypertension. Aging adults with low social support had higher mean BMI and mean SBP than adults with high social support.

### Associations of combinations of deficit in social supports with anthropometry, by gender

Figs 1 and 2 present the adjusted means (prediction mean derived from the fitted models) for BMI and WC, respectively, stratified by levels of social support (low vs. high) in men and women. These figures illustrate the association between combined social supports and variations in BMI and WC by gender.

Low availability of two types of support was associated with the highest levels of BMI among women (Fig 1). Women with low informational support and low tangible support showed the greatest difference in adjusted mean BMI (27.95 kg/m$^2$ [27.93, 27.97]) compared to women with high informational support and high tangible support (27.34 kg/m$^2$ [27.30, 27.38]) (S5 Table in S1 File). All low-low combinations of support were linked to the highest adjusted mean BMI levels among women (range: 27.92 to 27.95 kg/m$^2$). There was a linear trend observed for the combination of informational support with each of the three other types of support, suggesting that the absence of either informational support or another support had similar adverse effects on BMI among women. The joint effect of lack of two supports on BMI was less clear among men. Specific combinations of high belonging support or high emotional support with low availability of a second type of support were associated with the highest adjusted mean BMI levels in the range of 28.45 to 28.64 kg/m$^2$; that is, most low-low, or even high-high, combinations of social supports were not associated with, respectively, the highest or lowest adjusted mean BMI levels among men (Fig 1 and S5 Table in S1 File).

Similar gender-specific, but much weaker, associations between two social supports were seen for adjusted mean levels of WC among women and men (Fig 2 and S5 Table in S1 File). Again, all low-low combinations were associated with the highest adjusted mean WC levels among women (range: 88.62 to 88.71 cm), whereas all high-high combinations of social supports were associated with the lowest adjusted mean WC levels among women (range: 86.96 to 87.06 cm). The greatest difference in adjusted mean WC was seen among women with low informational support and low emotional support (88.69 cm [88.62, 88.76]) compared to women with high informational and high emotional supports (86.88 cm [86.75, 87.01]). Similar to BMI, a lack of either informational support or another support had similarly adverse effects on WC among women. By contrast, men showed a mostly flat pattern of association between different low-high combinations of two social supports and mean WC with average WC levels for all high-high combinations ranging from 99.73 cm to 99.99 cm and all low-low combinations ranging from 100.02 cm to 100.32 cm. Among men, the largest difference in anthropometric measures was seen for the combination of informational and tangible support.

**Table 1. Descriptive characteristics across four types of social support in the CLSA cohort at baseline (2011−15).**

| | N | Women | Mean (SD) age | Highest education | Non-smoker | Mean (SD) WC | Central obesity | Mean (SD) BMI | General obesity | Mean (SD) SBP | Mean (SD) DBP | Hyper-tension |
|---|---|---|---|---|---|---|---|---|---|---|---|---|
| **Total** | 28,779 | 14627 (51%) | 62.73 (10.18) | 13123 (46%) | 9107 (32%) | 94.12 (14.59) | 12630 (44%) | 28.03 (5.37) | 8404 (29%) | 120.98 (16.63) | 73.87 (9.98) | 14171 (49%) |
| **Informational support** | | | | | | | | | | | | |
| High (20) | 7650 | 4024 (53%) | 61.58 (9.87) | 4014 (52%) | 2681 (35%) | 93.13 (14.38) | 3143 (41%) | 27.78 (5.20) | 2084 (27%) | 120.20 (16.31) | 73.99 (9.68) | 3454 (45%) |
| Low (4–19) | 21,129 | 10603 (50%) | 63.15 (10.26) | 9109 (43%) | 6426 (30%) | 94.48 (14.65) | 9487 (45%) | 28.12 (5.42) | 6320 (30%) | 121.26 (16.74) | 73.83 (10.09) | 10717 (51%) |
| **Tangible support** | | | | | | | | | | | | |
| High (20) | 8685 | 3827 (44%) | 62.47 (9.81) | 4517 (52%) | 2884 (33%) | 94.52 (14.29) | 3636 (42%) | 27.92 (5.11) | 2391 (28%) | 120.68 (16.18) | 73.97 (9.71) | 4190 (48%) |
| Low (4–19) | 20,094 | 10800 (54%) | 62.85 (10.34) | 8606 (43%) | 6223 (31%) | 93.95 (14.72) | 8994 (45%) | 28.07 (5.47) | 6013 (30%) | 121.11 (16.82) | 73.83 (10.10) | 9981 (50%) |
| **Emotional support** | | | | | | | | | | | | |
| High (35) | 7125 | 3552 (50%) | 61.57 (9.88) | 3628 (51%) | 2442 (34%) | 93.61 (14.45) | 2943 (41%) | 27.85 (5.18) | 1980 (28%) | 120.25 (16.23) | 74.02 (9.74) | 3238 (45%) |
| Low (7–34) | 21,654 | 11075 (51%) | 63.12 (10.25) | 9495 (44%) | 6665 (31%) | 94.29 (14.63) | 9687 (45%) | 28.08 (5.42) | 6424 (30%) | 121.22 (16.75) | 73.83 (10.06) | 10933 (50%) |
| **Belonging support** | | | | | | | | | | | | |
| High (20) | 8026 | 3913 (49%) | 61.92 (9.82) | 4003 (50%) | 2643 (33%) | 93.90 (14.38) | 3365 (42%) | 27.93 (5.16) | 2255 (28%) | 120.65 (16.21) | 74.06 (9.68) | 3744 (47%) |
| Low (4–19) | 20,753 | 10714 (52%) | 63.05 (10.30) | 9120 (44%) | 6464 (31%) | 94.20 (14.67) | 9265 (45%) | 28.06 (5.44) | 6149 (30%) | 121.11 (16.79) | 73.80 (10.09) | 10427 (50%) |

DBP, diastolic blood pressure; SBP, systolic blood pressure. Hypertension was determined based on SBP ≥ 140 mm Hg and/or DBP ≥ 90 mm Hg (except patients with diabetes, ≥ 130/80), self-reported diagnosis of hypertension, or use of medication for hypertension. Highest education level: minimum having a Bachelor's degree from a university, or higher.

Combinations of social support were not associated with the odds of obesity in either gender. However, men with low informational support had 21% higher odds of general obesity (OR 1.21 [1.06, 1.38]) and 16% higher odds of central obesity (1.16 [1.01, 1.34]) (S6 Table in S1 File). Sensitivity analyses with other potential confounders showed similar adjusted mean levels of BMI and WC as the main results (S9–S12 Tables in S1 File).

### Associations of combinations of deficit in social supports with blood pressure, by gender

Figs 3 and 4 present the adjusted means (prediction mean derived from the fitted models) for SBP and DBP, respectively, stratified by levels of social support (low vs. high) and gender. Specific social support combinations were associated with adjusted mean SBP differentially for women and men (Fig 3). Among women, low availability of informational support, with or without deficits in a second support type, was associated with the highest adjusted mean SBP levels (range: 119.94 to 119.95 mmHg) (S7 Table in S1 File). Only the combination of informational support and emotional support appeared to be linearly associated with adjusted mean SBP among women: that is, women with high informational and emotional support had the lowest adjusted mean SBP level (118.2 mmHg [117.96, 118.44]) than all the other high-high combinations of social support (range: 118.48 to 118.56 mmHg). Notably, higher levels of SBP were also seen among women with mixed availability of social supports, such as high belonging support combined with low emotional support (119.77 [119.41, 120.13]), high tangible support with low emotional support (119.88 [119.56, 120.19]), and high tangible support with low belonging support (120.13 [119.79, 120.48]). Among men, low-low combinations of two social supports were not consistently

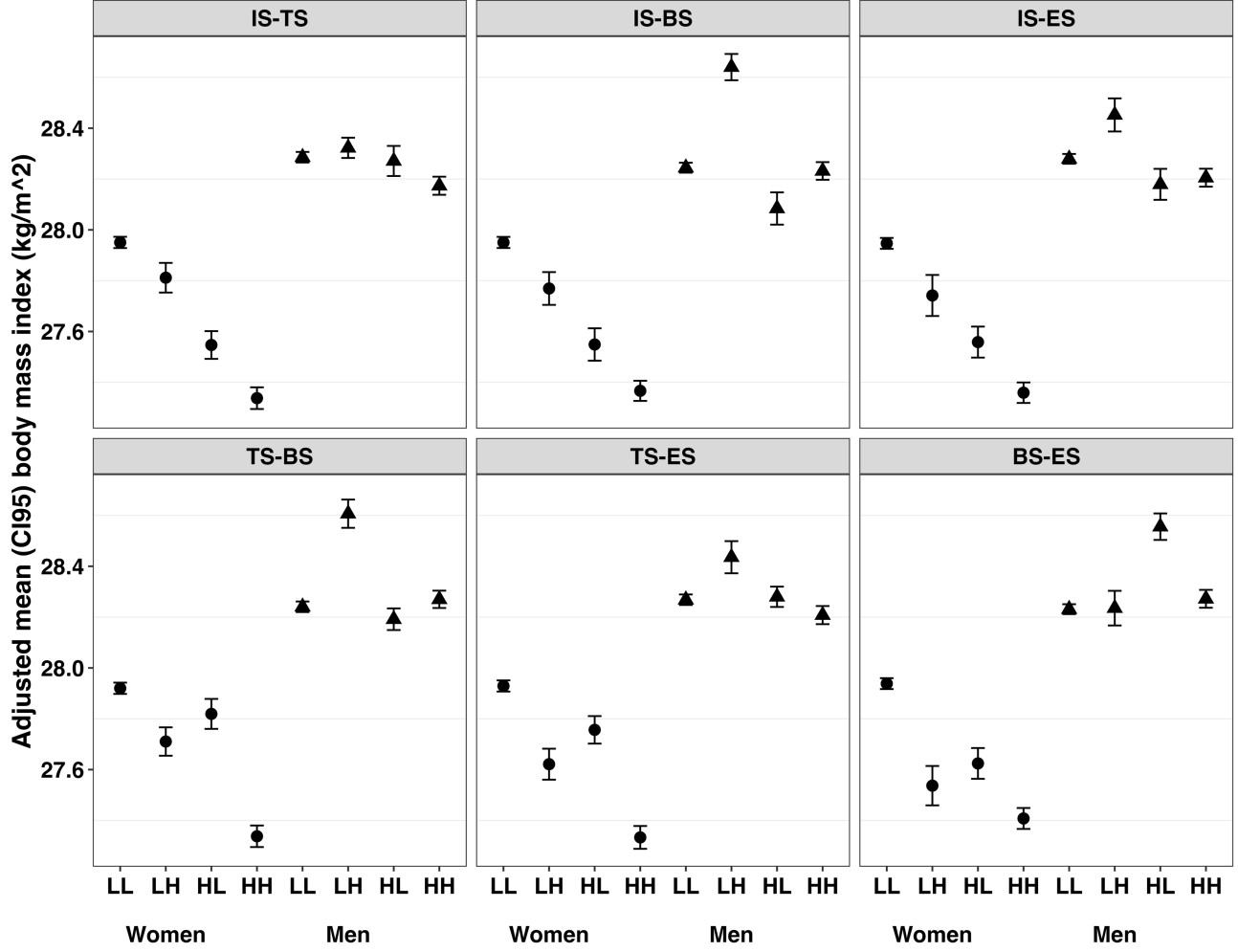

**Fig 1. Adjusted mean BMI associated with combinations of deficit in four types of perceived social support among women and men.** Panel A, tangible and informational support; Panel B, belonging and informational support; Panel C, emotional and informational support; Panel D, belonging and tangible support; Panel E, emotional and tangible support; Panel F, belonging and emotional support. IS, informational support; TS, tangible support; BS, belonging support; ES, emotional support; H, high support; L, low support.

associated with the highest adjusted mean SBP levels, and only the combination of tangible and emotional support appeared to have a graded association with SBP. The highest mean SBP among men was observed for the combination of high belonging support and low tangible support (123.18 mmHg [122.95, 123.4]), while the lowest mean SBP among men was seen for the combination of low belonging support and high emotional support (121.40 mmHg [121.13, 121.67]).

Adjusted mean DBP levels appeared more similar between women and men for four of the six different combinations of social supports (Fig 4). Adjusted mean DBP levels were lowest when high tangible support combined with low informational, belonging or emotional support. The highest levels of DBP among women were observed for those reporting high emotional support combined with low tangible support (72.25 mmHg [72.15, 72.34]), low informational support (72.23 mmHg [72.1, 72.36]) and low belonging support (72.21 mmHg [72.09, 72.33]). Among men, the highest DBP levels were seen for those reporting high informational support and low tangible support (76.72 mmHg [76.5, 76.94]).

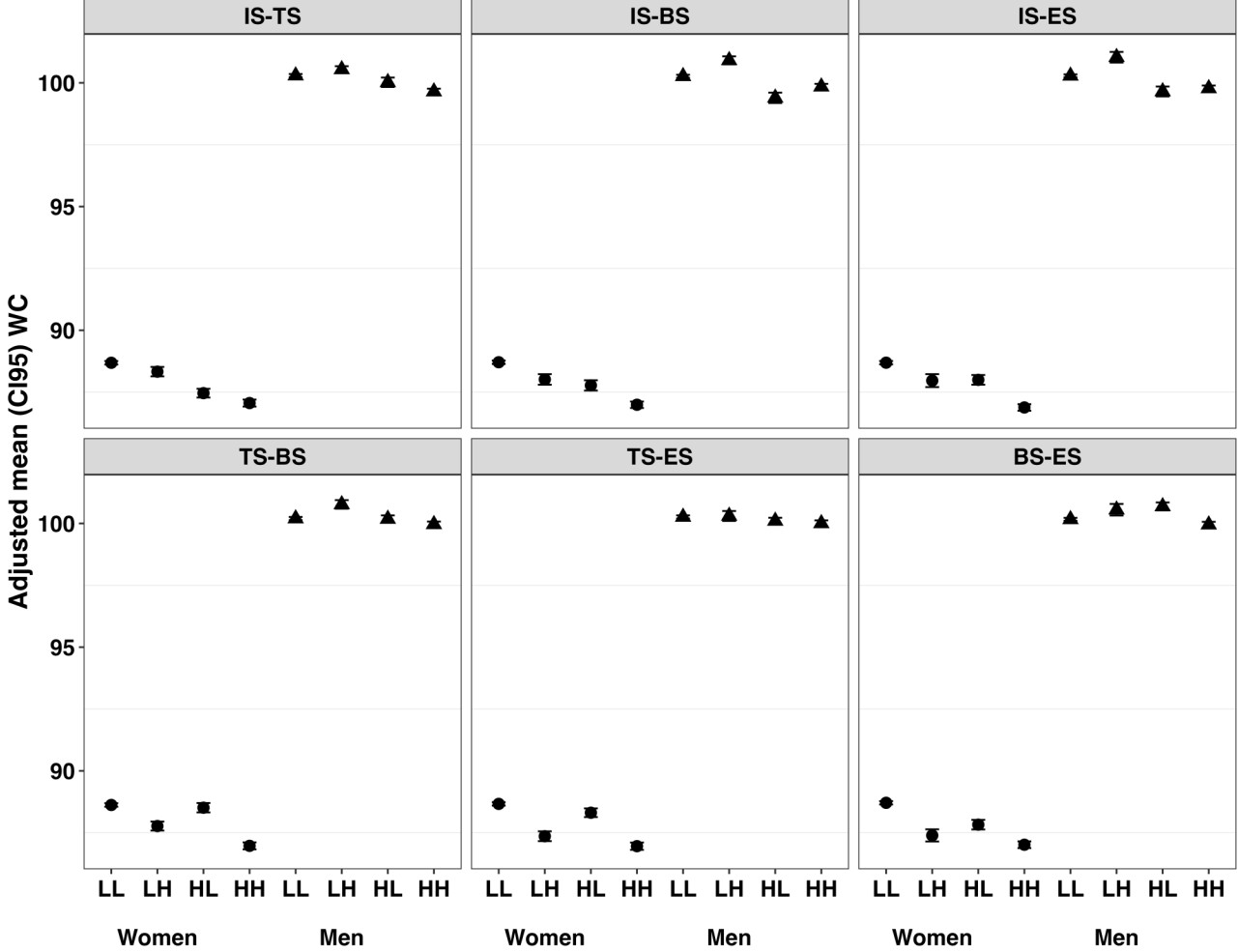

**Fig 2. Adjusted mean WC associated with combinations of deficit in four types of perceived social support among women and men.** Panel A, tangible and informational support; Panel B, belonging and informational support; Panel C, emotional and informational support; Panel D, belonging and tangible support; Panel E, emotional and tangible support; Panel F, belonging and emotional support. IS, informational support; TS, tangible support; BS, belonging support; ES, emotional support; H, high support; L, low support.

Combinations of social support were not associated with the odds of hypertension among women or men (S8 Table in S1 File). Sensitivity analyses with other potential confounders showed similar adjusted mean levels of blood pressure as the main results (S13–S16 Tables in S1 File).

## Discussion

This population-based, cross-sectional study of a large aging cohort in Canada comprehensively examined multiple types of social support that are typically conflated in the literature and revealed that specific combinations of dual deficits were associated with CVRFs in a gendered way. This study is novel in addressing gender equity in CVD prevention research and in offering new and contradicting insights on the assumed synergistic/multiplying effects of double deficits in upstream determinants of health. It demonstrates that average anthropometric and blood pressure measures were much higher for each type of low social support among women when they also lacked a second support type. For example, all low-low

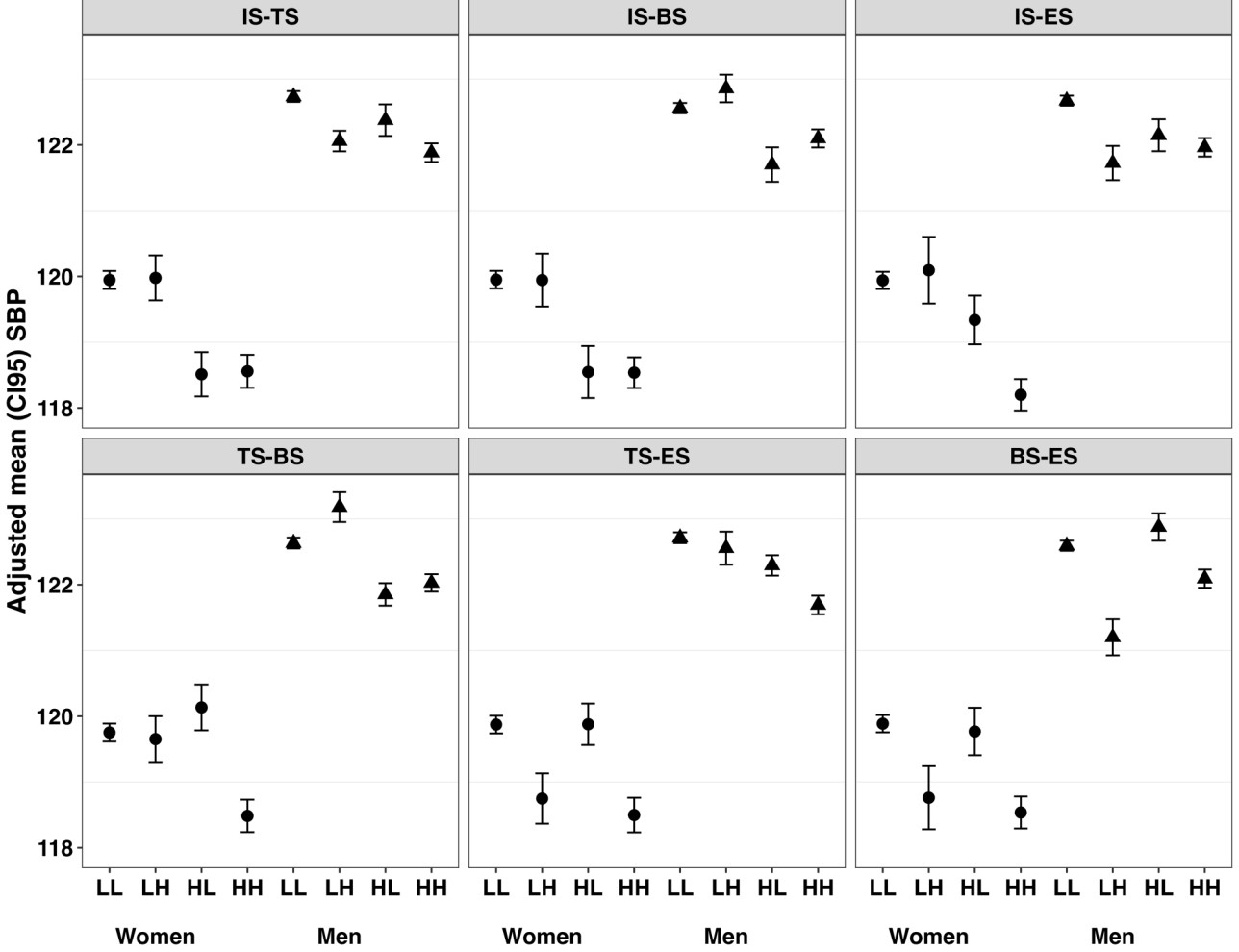

**Fig 3. Adjusted mean SBP associated with combinations of deficit in four types of perceived social support among women and men.** Panel A, tangible and informational support; Panel B, belonging and informational support; Panel C, emotional and informational support; Panel D, belonging and tangible support; Panel E, emotional and tangible support; Panel F, belonging and emotional support. IS, informational support; TS, tangible support; BS, belonging support; ES, emotional support; H, high support; L, low support.

combinations of social supports were associated with adjusted mean WC levels more than 88 cm that correspond to 'very high risk' for heart disease among women. In addition, low availability of informational support, with or without deficits in a second support type, was associated with the highest adjusted mean SBP levels among women. Only informational support alone was associated with obesity among men. Findings did not, however, reveal an antagonistic/synergistic effect—one type of social support did not exacerbate/mitigate the negative effect of another on CVRFs as others have postulated. Overall, results indicated that women and men differed quite dramatically in the specific configurations of social support deficits that were most strongly associated with major CVRFs.

### Findings in the context of previous research

Both perceived and received social support can influence CVRFs in older adults [8,28,29], and thus social support is viewed as essential to comprehensive efforts to mitigate CVD burden [28]. But, there is a need to examine social support as a

**Fig 4. Adjusted mean DBP associated with combinations of deficit in four types of perceived social support among women and men.** Panel A, tangible and informational support; Panel B, belonging and informational support; Panel C, emotional and informational support; Panel D, belonging and tangible support; Panel E, emotional and tangible support; Panel F, belonging and emotional support. IS, informational support; TS, tangible support; BS, belonging support; ES, emotional support; H, high support; L, low support.

multidimensional concept relevant to CVD risk prevention. [8] Emotional support in particular is identified as one important dimension of support for physiological processes of cardiovascular, endocrine and immune systems; [8] emotional/informational support are also associated with incident mild cognitive impairment in older women [30]. Informational support from neighbors and emotional support from close friends were each found to interact with feelings of loneliness in predicting hypertension among rural Chinese middle-age and older adults [31]. Four types of support (informational, emotional, belonging and tangible) were each associated with hypertension among Canadian older adults, with more consistent associations among women [14]. Similarly, all four types of social support were associated with adiposity only among older Canadian women [15]. The potential interplay of different types of social support as a unique determinant of CVD risk among older adults has not yet been studied. This study of CVRFs among aging adults is the first, to our knowledge, to assess inter-relations of multiple social supports.

Limited research suggests that social support interacts with other psychosocial (e.g., optimism, social strain, stress) and/or socioeconomic resources (e.g., education, income) to influence cardiovascular-related health outcomes. A study

of a US birth cohort showed that social support was a positive psychosocial resource that predicted healthy BMI in adults and buffered adverse effects of childhood socioeconomic disadvantage [32]. Among adult partners, supportive marital relationships buffered the negative association between income and ambulatory DBP [33]. Joint effects of social support and education on blood pressure were also reported among African American adults: higher received social support was positively associated with higher SBP among individuals with low education levels [34]. In a small community sample of older adults, emotional support was a coping resource that both predicted depressive symptoms and also moderated the impact of stress on depressive symptoms [9]. Main effects, stress-buffering and joint effects were also observed for social support and psychological wellbeing in Mexican-origin adults, with high social support buffering against the negative association of life stress on psychological wellbeing [11]. Although most research indicates that social support and other social-level resources have a joint effect on cardiovascular health, other studies show that the moderating role of social support in hypertension may not buffer social stressors in all settings [31]. Previous reports of amplifying synergistic effects on health outcomes do not mirror this study's findings of higher mean CVRFs that appeared additive for women with low social support when they also lacked a second type of social support.

Our results indicated that women and men were differentially vulnerable to greater CVD risk from distinct combinations of poor social supports. Overall, findings demonstrated that women fared worse in their CVD risk profiles in the absence of both informational support and a second type of social support. This finding was more notable for BMI and WC than for SBP. By contrast, among men, there was only one clear association between two absent social supports and CVRFs. The direct effect of social support and its interactions with other factors on BP has been shown to differ for women than men. In a small community study of about 200 adults aged 21–50 years, social support was examined in relation to blunted nocturnal dipping as an independent predictor of cardiovascular morbidity and mortality; results showed that normotensive adults with low levels of social support had blunted dipping with greater benefits of social support seen among women than among men [35]. Another study of 2348 married/co-habiting adults aged 25–75 years showed that supportive networks could buffer the insalubrious effects of strained interactions on wellbeing and health outcomes, but the buffering role of friends and family was observed more frequently among women [12]. Although the direct effects of social support on cardiovascular-related outcomes appears stronger or more consistent among women adults across the age spectrum, the effects of social support on cardiovascular-related behaviours may be stronger for men. Recent research from Spain showed that middle-age and older men who lack social support have the lowest adherence to cardiovascular screening (e.g., self-reported testing of blood pressure) and lifestyle recommendations whereas older women have high adherence irrespective of social support [36]. Thus, it is clear that women and men differ in the relationship between singular or joint social supports and CVD risk, with the interplay of dual deficits in social support seemingly additive in their association with objectively measured CVRFs among older women in Canada.

## Strengths and limitations

Although this study is cross-sectional and thus prohibits any causal inference, it provides information on which associations to examine prospectively. Self-reported measures of perceived availability of social support can be subject to recall or survey effect bias, however multiple outcomes were clinically assessed. Results may be biased due to unmeasured confounding from lipids which are also an important CVRF but blood sample data were not available for our data request. Findings may also be limited by residual confounding from imprecise measurement of self-reported covariables such as smoking or other health behaviours that could introduce bias that would either attenuate or inflate observed associations [37]; these factors could also be potential mechanisms underlying the association of social support and CVRFs. Finally, our study's external validity is limited by the nature of the CLSA cohort that is comprised of predominantly White, cisgender, and heterosexual adults. Nevertheless, our focus on women provided important disaggregated data that could reveal how the role of social support for CVRFs may be gendered from the lens of power relations between women and men

[38]. Finally, our results can only be generalized to middle-age and older adults living in the community in similar high-income countries.

There are multiple notable strengths of this population-based study which include: a large sample size, adjustment for multiple known confounders (including physical activity, FV intake and diabetes), sex-and-gender-based analysis, interaction effects, and objectively measured CVRFs. This study explicitly focused on joint associations of four types of social support instead of a summary index that masks the relative contribution of each support type. A particular strength is research attention to this more complete picture of social support and potential synergy that matter for women's heart health. This focus on gender and the multiple mutually reinforcing social supports thereby adds more precision to CVD prevention literature with implications for both research and clinical practice.

## Conclusions

This study demonstrated that a double burden of two deficits in social support showed much worse CVD risk profiles than a single deficit only for older women in Canada. Among men, only low informational support alone was associated with obesity. Overall, results contribute empirical knowledge on the unique joint associations between multiple social supports for CVRFs that suggest a more additive than an antagonistic interactive effect. Care and prevention of CVD for older women would benefit from addressing deficits in several types of social support, especially lack of informational support.

## Supporting information

**S1 File. Supplementary material.**
(DOCX)

## Acknowledgments

We thank the Canadian Longitudinal Study on Aging (CLSA) for providing the data that made this research possible. We also thank the reviewers for their constructive feedback. We acknowledge this research was performed by AS while at the University of British Columbia.

## Author contributions

**Conceptualization:** Annalijn I Conklin, Nadia A. Khan.

**Data curation:** Annalijn I Conklin, Abdollah Safari.

**Formal analysis:** Abdollah Safari.

**Funding acquisition:** Gerry Veenstra, Nadia A. Khan.

**Investigation:** Annalijn I Conklin, Abdollah Safari, Gerry Veenstra, Nadia A. Khan.

**Methodology:** Abdollah Safari.

**Project administration:** Annalijn I Conklin.

**Resources:** Annalijn I Conklin.

**Supervision:** Annalijn I Conklin, Gerry Veenstra, Nadia A. Khan.

**Validation:** Annalijn I Conklin.

**Visualization:** Abdollah Safari.

**Writing – original draft:** Annalijn I Conklin.

**Writing – review & editing:** Abdollah Safari, Gerry Veenstra, Nadia A. Khan.

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
