## [Decision Letter · Decision Letter 0]

Dear Dr. Conklin,

Thank you for submitting your manuscript to PLOS ONE. After careful consideration, we feel that it has merit but does not fully meet PLOS ONE’s publication criteria as it currently stands. Therefore, we invite you to submit a revised version of the manuscript that addresses the points raised during the review process.

We look forward to receiving your revised manuscript.

Kind regards,

Mohammad Reza Mahmoodi, Ph.D.

Academic Editor

PLOS ONE

**Journal Requirements:**

This research was funding by the Canadian Institutes of Health Research Catalyst Grant for Secondary Analysis of the CLSA (#162987)

"none declared"

Reviewers' comments:

Reviewer's Responses to Questions

**Comments to the Author**

1. Is the manuscript technically sound, and do the data support the conclusions?

Reviewer #1: Partly

Reviewer #2: No

2. Has the statistical analysis been performed appropriately and rigorously?

Reviewer #1: No

Reviewer #2: Yes

3. Have the authors made all data underlying the findings in their manuscript fully available?

Reviewer #1: No

Reviewer #2: Yes

4. Is the manuscript presented in an intelligible fashion and written in standard English?

Reviewer #1: Yes

Reviewer #2: No

**Reviewer #1:**  Title: OK.

Abstract: According to the expressed objectives, the method of exposure measurements related to social supports are not addressed in the method part of this section. Please show the results of the study numerically instead of narratively. In this case the conclusion will be sensible for the readers.

Keywords: Ok.

Introduction: Please show the novelty of the study in this section. It is also desirable to show the expected practical use of the results of the study in this section.

Material and method: Please notice to the following issues in this section:

- I think it is necessary to define the sampling method and setting of the main study of population-based Canadian Longitudinal Study on aging in more detail.

- Please state how non-response was managed in this study.

- I think it is essential to define how CVD related risk factors were measured.

- Please define the methods used for validating and reliability of the responses used for measuring variables used for assessing the social support. It is also essential to define these variables in detail.

- The definition of the variables shown under the section of “covariables” shall be defined. For instance, what exactly do you mean by physical activity or weekly alcohol intake?

- Please clarify the variables considered for forming the PCs. It is also essential to show the internal consistency of these variables as a proxy for validating this procedure.

- It seems the models, the results of which are shown in figures 2 and 3 are not addressed clearly in this section.

- According to the above-mentioned issues the following section of result and discussion are not justifiable.

**Reviewer #2:**  • A recent systematic review (as following) examined the effect of social support on cardiovascular risk. What was the research gap for the current study?

- Singh M, Nag A, Gupta L, Thomas J, Ravichandran R, Panjiyar BK. Impact of Social Support on Cardiovascular Risk Prediction Models: A Systematic Review. Cureus. 2023 Sep 24;15(9):e45836. doi: 10.7759/cureus.45836. PMID: 37881384; PMCID: PMC10597590.

• Among multiple risk factors of cardiovascular diseases, only two anthropometric measures (body mass index and waist circumference) in addition to systolic and diastolic blood pressure have been considered as the indicators of cardiovascular disease (CVD) risk. Some other important factors such as physical activity, nutrition pattern, blood glucose and lipid profile were not considered. They should be mentioned in limitations of the study. Also, these factors might change the association between social support and cardiovascular risk.

• In “Results” of “Abstract” section, it is recommended to provide more statistical findings.

• “Keywords” are recommended to initiate with capital letters.

• Why did the authors initiate the phrase of “Comprehensive cohort” with a capital letter in “Methods” section?

• Please provide a brief description of the four types of social support in “Introduction” section.

• Please check the sentences in Lines 170-175. The verbs should be written in past tense.

**Do you want your identity to be public for this peer review?** For information about this choice, including consent withdrawal, please see our Privacy Policy

Reviewer #1: **Yes: ** Babak Eshrati

Reviewer #2: No

---

## [Author Response · Author response to Decision Letter 1]

10 Dec 2024

We have uploaded a point-by-point response to each comment of Reviewer 1 and Reviewer 2. In addition, we have addressed the issues raised by the Editor within the revised manuscript.

---

## [Decision Letter · Decision Letter 1]

Dear Dr. Conklin,

Thank you for submitting your manuscript to PLOS ONE. After careful consideration, we feel that it has merit but does not fully meet PLOS ONE’s publication criteria as it currently stands. Therefore, we invite you to submit a revised version of the manuscript that addresses the points raised during the review process.

I hope you will response to peer reviewer comments and modify your manuscript again for possible publication. Incomplete responses and failure to correct the manuscript will certainly cause problems in the review and publication process.

We look forward to receiving your revised manuscript.

Kind regards,

Prof. Mohammad Reza Mahmoodi, Ph.D.

Academic Editor

PLOS ONE

Journal Requirements:

Reviewers' comments:

Reviewer's Responses to Questions

**Comments to the Author**

Reviewer #1: (No Response)

Reviewer #2: All comments have been addressed

2. Is the manuscript technically sound, and do the data support the conclusions?

Reviewer #1: Yes

Reviewer #2: Yes

3. Has the statistical analysis been performed appropriately and rigorously?

Reviewer #1: No

Reviewer #2: Yes

4. Have the authors made all data underlying the findings in their manuscript fully available?

Reviewer #1: Yes

Reviewer #2: Yes

5. Is the manuscript presented in an intelligible fashion and written in standard English?

Reviewer #1: Yes

Reviewer #2: Yes

Reviewer #1: this is a very important study in terms of the subject. however please notice to teh following comments:

Title: according to the proposed objective it is better to convey the title with a SMART format (e.g. “the association of social support with cardiovascular disease risk factors in middle age adults”).

Abstract: Please define the proposed dependent and independent variables explicitly in the material and method part of this section. On the other hand, please show the effect measures of the association of social support with the expressed risk factors. In this case the conclusion will be more sensible for the readers.

Keywords: OK

Introduction: I think it is necessary to express the novelty of the study explicitly in this section.

Material and method: Please notice to the following issues in this section:

- Please define the main study of CLSA in more detail (especially in terms of sampling method and measurement tools and reliability and validity of the variable measurements).

- In contrast to detailed description of the social support categories, it is not clear how different types have been measured.

- The method of additive and multiplicative association assessment shall be defined in more detail. On the other hand, it is necessary to principal component analysis in more detail so that it will be clarified which variables have been considered as the recruited assets of PCA.

- Please define the used sensitivity analysis in more detail.

- It is necessary to express how different variables were considered to be recruited for the multivariable analysis.

Results: part of this section is not justifiable according to the above-mentioned issues.

Discussion: Please express the limitations of the study in this section as well.

References: OK

Reviewer #2: Thanks for your answer to my recommendations. All the suggested points have been considered or replied by the authors.

**Do you want your identity to be public for this peer review?** For information about this choice, including consent withdrawal, please see our Privacy Policy

Reviewer #1: **Yes: ** Babak Eshrati

Reviewer #2: No

---

## [Author Response · Author response to Decision Letter 2]

10 Feb 2025

We have uploaded a point by point response to the remaining reviewer comments. Please see the attached document for details.

---

## [Decision Letter · Decision Letter 2]

Dear Dr. Conklin,

Thank you for submitting your manuscript to PLOS ONE. After careful consideration, we feel that it has merit but does not fully meet PLOS ONE’s publication criteria as it currently stands. Therefore, we invite you to submit a revised version of the manuscript that addresses the points raised during the review process.

We look forward to receiving your revised manuscript.

Kind regards,

Prof. Mohammad Reza Mahmoodi, Ph.D.

Academic Editor

PLOS ONE

Journal Requirements:

**Additional Editor Comments:**

Dear Corresponding Author

Kindly respond to the referee's comments and make the necessary modifications to the text of the manuscript for greater clarification and transparency. In addition, prepare and send a rebuttal letter for each of the referee's comments.

Reviewers' comments:

Reviewer's Responses to Questions

**Comments to the Author**

Reviewer #1: All comments have been addressed

2. Is the manuscript technically sound, and do the data support the conclusions?

Reviewer #1: Yes

3. Has the statistical analysis been performed appropriately and rigorously?

Reviewer #1: Yes

4. Have the authors made all data underlying the findings in their manuscript fully available?

Reviewer #1: Yes

5. Is the manuscript presented in an intelligible fashion and written in standard English?

Reviewer #1: Yes

Reviewer #1: thanks to distinguished authors, I think most of the comments are addressed. Please just notice to the following minor issues:

- I could not understand why just findings related to women are shown in the abstract.

- - The supplementary material was not included.

- Please the reliability indices used for the assessment of the responses social support variables.

- I think it is necessary to define the method of interaction (both additive and multiplicative ones) in more detail. (This is issue may be due to the fact that supplemental material was not included).

**Do you want your identity to be public for this peer review?** For information about this choice, including consent withdrawal, please see our Privacy Policy

Reviewer #1: **Yes: ** Babak Eshrati

---

## [Author Response · Author response to Decision Letter 3]

1 May 2025

We have uploaded a file describing our response to each comment of the reviewer.

---

## [Decision Letter · Decision Letter 3]

Cardiovascular risk in aging adults with double deficits in social support: a gender-sensitive, cross-sectional analysis of the CLSA cohort

PONE-D-24-33347R3

Dear Dr. Conklin,

We’re pleased to inform you that your manuscript has been judged scientifically suitable for publication and will be formally accepted for publication once it meets all outstanding technical requirements.

Kind regards,

Prof. Mohammad Reza Mahmoodi, Ph.D.

Academic Editor

PLOS ONE

Additional Editor Comments (optional):

Reviewers' comments:

Reviewer's Responses to Questions

**Comments to the Author**

Reviewer #1: (No Response)

2. Is the manuscript technically sound, and do the data support the conclusions?

Reviewer #1: (No Response)

3. Has the statistical analysis been performed appropriately and rigorously?

Reviewer #1: (No Response)

4. Have the authors made all data underlying the findings in their manuscript fully available?

Reviewer #1: (No Response)

5. Is the manuscript presented in an intelligible fashion and written in standard English?

Reviewer #1: (No Response)

Reviewer #1: thanks to the distinguished authors all of the previous comments are addressed in the letter and context of the paper.

**Do you want your identity to be public for this peer review?** For information about this choice, including consent withdrawal, please see our Privacy Policy

Reviewer #1: **Yes: ** Babak Eshrati

---

## [Editor Report · Acceptance letter]

PONE-D-24-33347R3

PLOS ONE

Dear Dr. Conklin,

I'm pleased to inform you that your manuscript has been deemed suitable for publication in PLOS ONE. Congratulations! Your manuscript is now being handed over to our production team.

Kind regards,

on behalf of

Professor Mohammad Reza Mahmoodi

Academic Editor

PLOS ONE